

# The efficacy of single mitochondrial genes at reconciling the complete mitogenome phylogeny—a case study on dwarf chameleons

Devon C. Main[1], Jody M. Taft[2,3], Anthony J. Geneva[4], Bettine Jansen van Vuuren[1] and Krystal A. Tolley[1,3]

[1] Centre for Ecological Genomics and Wildlife Conservation, University of Johannesburg, Johannesburg, Gauteng, South Africa

[2] School of Animal, Plant and Environmental Sciences, University of the Witwatersrand, Johannesburg, South Africa

[3] South African National Biodiversity Institute, Kirstenbosch Research Centre, Claremont, South Africa

[4] Department of Biology, Center for Computational and Integrative Biology, Rutgers, The State University of New Jersey, Camden, NJ, United States of America

Corresponding author
Krystal A. Tolley,
k.tolley@sanbi.org.za

## ABSTRACT

Although genome-scale data generation is becoming more tractable for phylogenetics, there are large quantities of single gene fragment data in public repositories and such data are still being generated. We therefore investigated whether single mitochondrial genes are suitable proxies for phylogenetic reconstruction as compared to the application of full mitogenomes. With near complete taxon sampling for the southern African dwarf chameleons (*Bradypodion*), we estimated and compared phylogenies for the complete mitogenome with topologies generated from individual mitochondrial genes and various combinations of these genes. Our results show that the topologies produced by single genes (*ND2*, *ND4*, *ND5*, *COI*, and *COIII*) were analogous to the complete mitogenome, suggesting that these genes may be reliable markers for generating mitochondrial phylogenies in lieu of generating entire mitogenomes. In contrast, the short fragment of *16S* commonly used in herpetological systematics, produced a topology quite dissimilar to the complete mitogenome and its concatenation with *ND2* weakened the resolution of *ND2*. We therefore recommend the avoidance of this *16S* fragment in future phylogenetic work.

## INTRODUCTION

Mitochondria, the subcellular organelles that enable eukaryotes to respire aerobically, likely began their evolution as unicellular organisms that made their way into the ancestral progenitor of eukaryotic cells *via* a process of endosymbiosis approximately 2.5 billion years ago (*Gray, 2017*; *Sagan, 1967*). Mitochondria possess a distinct haploid genome, independent of the nuclear genome. Among vertebrates, mitochondria are, for the most part, uniparentally inherited along the maternal line and rarely, if ever,

experience any recombination (*Bágeľová Poláková et al., 2021*; *Shtolz & Mishmar, 2019*). Compared to protein-coding genes in the nuclear genome, mitochondria experience fast, sometimes clock-like, substitution rates (*Avise et al., 1987*; *Konrad et al., 2017*). Furthermore, mitochondria are far more abundant in the cell than the nuclei within which nuclear genomic DNA is located (*Bollmann et al., 2021*; *Filograna et al., 2021*). The abundance, haploidy and fast mutation rate of mitochondria have made mitochondrial DNA an attractive target for phylogenetic reconstruction of eukaryotic life (*Hajibabaei et al., 2007*).

In contrast to the long-standing reliance on mitochondrial genes in phylogenetics, the rapid expansion of next-generation sequencing technologies has ushered in a new era characterised by mitogenome-scale data (*Hayden, 2014*; *Lewin et al., 2022*; *Muir et al., 2016*). Despite the pace with which these genome-scale approaches are being assimilated into phylogenetics, many researchers still do not have access to resources to produce and analyse mitogenome-scale data. This is particularly pertinent for researchers based in low- and middle-income countries where funding resources are scarce, but biodiversity is plentiful (*Helmy, Awad & Mosa, 2016*). Indeed, researchers globally continue to make use of single or multiple gene fragments for phylogenetics, typically sequenced *via* Sanger Sequencing technologies. The abundance of mitochondria in tissues results in high DNA yield, which can overcome instances where DNA has been degraded. The popularity of sequencing single mitochondrial gene fragments over the past two decades has meant that public sequence data repositories have amassed immense stockpiles of single gene fragment data (*Joly et al., 2014*). By 2016, GenBank (https://www.ncbi.nlm.nih.gov/genbank/) alone had accumulated sequence data from 1,125,514 accessioned individuals of tetrapods, and this number continues to grow (*Gratton et al., 2017*).

As the cost of sequencing drops and mitogenome-scale data become more widely available, some important and interesting questions emerge. In particular, can the stockpiles of single gene data generated over the past three decades still be utilised in phylogenetic analyses, or are these data inferior and obsolete? It is therefore necessary to assess whether single gene data can be used accurately to reflect the evolutionary history of the mitochondrion, and/or to what extent mitogenome-scale data are necessary. These questions are not simple to answer; some nuance is required when interrogating them. Of course, more data allows for greater confidence in the assessment of complex genealogies. This is not being disputed, but, if we triage funding resources, then mitogenome-scale data may not be cost effective in terms of financial trade-off, particularly when research funds are limited. Even if the cost of sequencing drops to a point where mitogenome-scale data are accessible to all researchers worldwide, if single gene data are reliable proxies for mitogenome-scale data, then the democratisation of mitogenome-scale data should not render previously generated single gene data obsolete.

Somewhat surprisingly, whether single mito-gene data can accurately reconcile a mitogenome inferred phylogeny has rarely been investigated, with examples limited to cetaceans, and metazoans more generally (*Duchêne et al., 2011*; *Havird & Santos, 2014*). In both cases, certain single mito-genes and gene combinations were able to reconcile the mitogenome phylogeny. The most reliable genes for cetaceans, however, were not the same

as for metazoans. This suggests that the phylogenetic informativeness of mito-genes may be, at the least, broadly taxon-specific. It should be noted that even the complete mitogenome can only be used to infer the genealogy of the mitochondrion, but this genealogy may not necessarily reflect that of the taxon. For example, discordance between the mitochondrial and nuclear phylogenies has been documented for many taxa, and has been attributed to incomplete lineage sorting, assortative mating, asymmetric introgression and/or sex-linked selection (*Després, 2019*; *Fossøy et al., 2016*; *Tóth et al., 2017*; *Wendt et al., 2022*). Thus, keeping in mind the potential for discordance between genomes is key for studies that incorporate mitochondrial DNA.

Despite the increased availability of mitogenome-scale data, studies comparing single gene phylogenies to the mitogenome phylogeny are notably lacking for reptiles—a group for which there exists an abundance of single gene data in publicly available repositories. For example, southern African dwarf chameleons of genus *Bradypodion* have largely been the subject of single locus (concatenated mito-genes) phylogenetics (*Tolley et al., 2004*; *Tolley, Chase & Forest, 2008*; *Tolley, Tilbury & Burger, 2022*). As a result of this, single mito-gene data are widely available for the genus on public data repositories. Most relationships between *Bradypodion* and other chameleon genera and most deep nodes within *Bradypodion* have been resolved through multi-locus phylogenetics that include mito-genes and nuclear genes (*Tolley, Townsend & Vences, 2013*). However, there are some nodes within the genus that are not yet well-resolved. Given the abundance of publicly available single gene fragment sequence data for *Bradypodion*, we used *Bradypodion* as a case study to evaluate the efficacy of single mitochondrial gene fragments as proxies in reconstructing the evolutionary history of the complete mitochondrial genome. With near complete taxon sampling for the genus, we contrasted topologies generated from single and concatenated mito-gene datasets to the full mitogenome topology. In particular, we hypothesised that the individual mito-gene topologies might show differing topologies amongst them as compared to the mitogenome, and that the individual gene trees would have lower node support than the topology generated by the mitogenome.

## MATERIALS & METHODS

### Mitogenome generation

Our sampling protocol followed the ARROW guidelines (*Field et al., 2019*). Specifically, we acquired institutional animal care approval from South African National Biodiversity Institute: 003/2011, 001/2013, 001/2014, 0001/2015, University of Johannesburg: 2019-10-10, and University of the Witwatersrand: 2019/10/56/B  for DNA sampling. We complied with relevant national, international and institutional regulations regarding animal care during DNA sampling, ensuring animals were in our care for minimal duration and under minimal stress. We took all measures possible to follow the 3R tenets, and taxon-specific guidelines for the genetic sampling of dwarf chameleons (*Herrel et al., 2012*). Total genomic DNA was extracted from tissue samples (tail tips) from dwarf chameleon individuals ($n = 44$) using the Qiagen DNeasy blood and tissue kit (Qiagen, Hilden, Germany). Samples were stored in either 99% ethanol or RNAlater and were collected during multiple

field trips in South Africa between 2010 and 2022 under permits from the provinces of Eastern Cape, KwaZulu-Natal, Limpopo, Mpumalanga, Northern Cape, Western Cape (018-CPM403-00001, 0092-CPM401-00006, AAA004-00107-0035, CPM-333-00002, CRO 3/19CR, CRO 35/15CR, CRO 36/15CR, CRO 32/20CR, CRO 33/20CR, FAUNA 110/2011, OP 4758/2010, OP 4596/2010, OP 2635, OP 1259/2014, KZN 1647/2009, MPB.5299, MPB.5371, MPB.5544, MPB.5604, CN44-59-5795, RSH 24/2021, WRO 41/03WR; WRO 15/03WR). Subsamples were exported to the USA under CITES regulations (permits # 142667, 206199, 257511, 260568 and 260570 issued by the South African Department of Fisheries, Forestry and Environment), with remaining tissue samples accessioned at the South African National Wildlife Biobank, Pretoria. Sampling included 14 of the 20 described species and three additional populations of taxonomic rank not yet established (Table 1). Genomic libraries were prepared using 10 ng of DNA per sample and an Illumina DNA Library Prep Kit which made use of Transposase-mediated Tagmentation. Samples were uniquely barcoded using IDT Nextera for Illumina DNA Unique Dual 10 bp Indexes adaptors. After library preparation, Illumina re-sequencing was carried out on an Illumina NovaSeq S4. Raw sequencing reads were trimmed, and adaptors were removed using trimmomatic v0.39 (*Bolger, Lohse & Usadel, 2014*) according to the following settings: LEADING:20 TRAILING:20 SLIDINGWINDOW:13:20 MINLEN:23 with the adaptor set to NexteraPE-PE.fa:2:30:10:4 and a minimum phred score of 33. Trimmed reads were then passed through NOVOPlasty v4.3.1 (*Dierckxsens, Mardulyn & Smits, 2017*) using a publicly available *Chamaeleo chamaeleon* complete mitogenome (accession number: NC012427; *Macey et al., 2008*) as starting seed and reference genome. Assembled mitogenomes were then annotated using the MITOS v1 web server (*Bernt et al., 2013*).

## Phylogenetic analyses

Once annotated, mitogenomes were imported into Geneious Prime 2022 (https://www.geneious.com) and published complete mitogenomes for seven other chameleon genera were downloaded from GenBank as outgroup taxa (Table 2). Using these 51 individuals (44 ingroup, seven outgroup), we created a total of 22 different alignments for subsequent phylogenetic analyses: a full mitogenome alignment (excluding the highly variable tRNAs which are difficult to align), 15 single locus alignments (13 protein-coding genes, two non-coding ribosomal genes), the short fragment of *16S* (*Alrefaei, 2022*; *Bates et al., 2013*; *Hay et al., 1995*; *Hertwig, de Sá & Haas, 2004*; *Lamb & Bauer, 2002*; *Main, Van Vuuren & Tolley, 2018*; *Main et al., 2022*; *Vences et al., 2004*; *Vences et al., 2005*; *Welton et al., 2010*), a commonly used short fragment of *COI* (*Hebert et al., 2003*), a concatenation of short *16S* fragment with *ND2* (*Tolley, Chase & Forest, 2008*; *Tolley, Tilbury & Burger, 2022*; as these two genes are regularly concatenated in the herpetological phylogenetic literature), a concatenation of *ND2* and *ND5* (shown by our analyses to be among the most reliable genes for phylogenetic reconstruction–see Results), a concatenation of both ribosomal subunits, and a concatenation of all protein-coding genes (PCGs). All new sequences from this study were deposited in GenBank (Table S1).

Maximum likelihood phylogenetic analyses were run for the complete mitogenome in IQ-TREE v2.0.3 (*Minh et al., 2020b*) with the optimal model of molecular evolution

**Table 1  List of species used to generate mitogenomes for the study, with sample number and collection locality.** See Table S1 for GenBank accession numbers.

| Species | | ID | Locality | Lat | Lon | Voucher † |
|---|---|---|---|---|---|---|
| *Bradypodion* | *barbatulum* | FP527 | Southern slopes of Kouga Mountains, Eastern Cape, South Africa | −33.74 | 23.71 | |
| *Bradypodion* | *caeruleogula* | CT359 | Ngoya Forest, KwaZulu-Natal, South Africa | −28.83 | 31.73 | |
| *Bradypodion* | *caffrum* | Cham503 | Hluleka Nature Reserve, Eastern Cape, South Africa | −32.82 | 29.30 | PEM R22146 |
| *Bradypodion* | *caffrum* | Cham509 | Hluleka Nature Reserve, Eastern Cape, South Africa | −32.82 | 29.30 | PEM R22147 |
| *Bradypodion* | sp. 3 | JdC10-026 | Greytown, KwaZulu-Natal, South Africa | −29.06 | 30.58 | |
| *Bradypodion* | *damaranum* | P306B | George, Western Cape, South Africa | −33.95 | 22.45 | |
| *Bradypodion* | *damaranum* | P783 | George, Western Cape, South Africa | −33.95 | 22.46 | |
| *Bradypodion* | *damaranum* | P822 | Idile Forest, Western Cape, South Africa | −33.95 | 22.56 | |
| *Bradypodion* | *damaranum* | P824 | Idile Forest, Western Cape, South Africa | −33.95 | 22.56 | |
| *Bradypodion* | *gutturale* | KTH08-51 | Ladismith Nature Reserve, Western Cape, South Africa | −33.53 | 21.24 | |
| *Bradypodion* | *melanocephalum* | H1390 | Westville, KwaZulu-Natal, South Africa | −29.35 | 29.99 | |
| *Bradypodion* | *melanocephalum* | P625 | Roosfontein Nature Reserve, KwaZulu-Natal, South Africa | −29.83 | 30.95 | |
| *Bradypodion* | *melanocephalum* | P657 | Roosfontein Nature Reserve, KwaZulu-Natal, South Africa | −29.86 | 30.93 | |
| *Bradypodion* | *melanocephalum* | P684 | Westville, KwaZulu-Natal, South Africa | −29.86 | 30.93 | |
| *Bradypodion* | *ngomeense* | FP596B | Ngome Forest, KwaZulu-Natal, South Africa | −27.82 | 31.42 | |
| *Bradypodion* | *occidentale* | KTH11-11 | Noup, Northern Cape, South Africa | −30.14 | 17.21 | |
| *Bradypodion* | *pumilum* | KTH288 | Durbanville, Western Cape, South Africa | −33.83 | 18.62 | |
| *Bradypodion* | *setaroi* | P457 | St Lucia, KwaZulu-Natal, South Africa | −28.37 | 32.41 | |
| *Bradypodion* | *setaroi* | H1273 | St Lucia, KwaZulu-Natal, South Africa | −28.38 | 32.41 | |
| *Bradypodion* | sp. 1 | FP578A | Kamberg, KwaZulu-Natal, South Africa | −29.38 | 29.66 | |
| *Bradypodion* | sp. 1 | R531A | Kamberg, KwaZulu-Natal, South Africa | −29.38 | 29.66 | |
| *Bradypodion* | sp. 2 | KTH437 | Karkloof Forest, KwaZulu-Natal, South Africa | −29.27 | 30.34 | SAM ZR51950 |
| *Bradypodion* | sp. 2 | KTH450 | Gilboa Forest (Plantation), KwaZulu-Natal, South Africa | −29.20 | 30.39 | SAM ZR51952 |
| *Bradypodion* | *taeniabronchum* | KTH108 | Tsitsikamma Mountains, Eastern Cape, South Africa | −33.93 | 24.14 | |
| *Bradypodion* | *taeniabronchum* | MBUR02439 | Thyspunt, Eastern Cape, South Africa | −34.17 | 24.72 | PEM R18762 |
| *Bradypodion* | *thamnobates* | P474 | Howick, KwaZulu-Natal, South Africa | −29.36 | 30.13 | |
| *Bradypodion* | *thamnobates* | P582 | Howick, KwaZulu-Natal, South Africa | −29.86 | 30.93 | |
| *Bradypodion* | *thamnobates* | P593 | Howick, KwaZulu-Natal, South Africa | −29.83 | 30.94 | |
| *Bradypodion* | *thamnobates* | TD52 | Dargle Valley, KwaZulu-Natal, South Africa | −29.49 | 30.06 | |
| *Bradypodion* | *thamnobates* | JDC09-134 | Boston, KwaZulu-Natal, South Africa | −29.69 | 30.07 | |
| *Bradypodion* | *transvaalense* | AMDSF219 | Graskop, Mpumalanga, South Africa | −24.96 | 30.79 | |
| *Bradypodion* | *transvaalense* | ATENTBT1 | Soutpansberg, Entabeni, Limpopo, South Africa | −22.99 | 30.28 | |
| *Bradypodion* | *transvaalense* | CT63 | Hendriksdal off R37, Mpumalanga, South Africa | −25.15 | 23.83 | PEM R5688 |
**Table 1** (*continued*)

| Species | | ID | Locality | Lat | Lon | Voucher † |
|---|---|---|---|---|---|---|
| *Bradypodion* | *transvaalense* | KTH06-03 | Soutpansberg, Lajuma, Limpopo, South Africa | −23.00 | 29.87 | PEM R17536 |
| *Bradypodion* | *transvaalense* | KTH06-04 | Soutpansberg, Lajuma, Limpopo, South Africa | −23.00 | 29.87 | PEM R17531 |
| *Bradypodion* | *transvaalense* | KTH533 | Barberton, Mpumalanga, South Africa | −25.80 | 31.11 | PEM R17527 |
| *Bradypodion* | *transvaalense* | KTH524 | Mt Sheba Forest, Mpumalanga, South Africa | −24.94 | 30.71 | PEM R17523 |
| *Bradypodion* | *transvaalense* | KTH529 | Mt Sheba Forest, Mpumalanga, South Africa | −24.94 | 30.71 | PEM R17522 |
| *Bradypodion* | *ventrale* | KTH156 | Groendal Nature Reserve, Eastern Cape | −33.71 | 25.31 | |
| *Bradypodion* | *ventrale* | N286 | Jeffreys Bay, Eastern Cape, South Africa | −33.98 | 24.96 | |
| *Bradypodion* | *ventrale* | N287 | Jeffreys Bay, Eastern Cape, South Africa | −33.98 | 24.96 | |
| *Bradypodion* | *ventrale* | T302 | Jeffreys Bay, Eastern Cape, South Africa | −33.98 | 24.96 | |
| *Bradypodion* | *venustum* | HB361 | Grootvadersbosch, Western Cape, South Africa | −33.99 | 20.81 | PEM R026336 |
| *Bradypodion* | *venustum* | N087 | Grootvadersbosch, Western Cape, South Africa | −33.98 | 20.83 | |

**Notes.**
† PEM, Port Elizabeth Museum (Bayworld); SAM, South African Museum.

**Table 2** Outgroup taxa used in the phylogenetic analyses for *Bradypodion*, associated GenBank accession numbers and studies in which they were first published.

| Species | | GenBank accession number | Study |
|---|---|---|---|
| *Brookesia* | *decaryi* | AB474914 | *Okajima & Kumazawa (2010)* |
| *Calumma* | *parsonii* | AB474915 | *Okajima & Kumazawa (2010)* |
| *Chamaeleo* | *dilepis* | EF222189 | *Macey et al. (1997)* |
| *Furcifer* | *oustaleti* | NC008777 | *Kumazawa (2007)* |
| *Kinyongia* | *fischeri* | EF222188 | Macey et al. (1997) |
| *Rieppeleon* | *kerstenii* | AB474918 | *Okajima & Kumazawa (2010)* |
| *Trioceros* | *melleri* | AB474916 | *Okajima & Kumazawa (2010)* |

estimated for each gene separately using the built-in ModelFinder implemented in IQ-TREE (*Chernomor, von Haeseler & Minh, 2016*; *Kalyaanamoorthy et al., 2017*). To assess whether node support would be improved by merging models of evolution for similarly evolving genes into larger partitions, a merged partitioned analysis was also run. This analysis makes use of the PartitionFinder (*Lanfear et al., 2017*) algorithm to generate a partitioning scheme within IQ-TREE and merge genes for model estimation to assess whether this reduces over-parameterization.

For the Bayesian analyses, we estimated the best evolutionary models and partitions for the complete mitogenome in PartitionFinder v2.1.1 (*Lanfear et al., 2017*). The output suggested that the dataset be partitioned into 10 partitions with GTR + *I* + *G* the optimal model of evolution for each partition (Table 3).

Bayesian phylogenetic analyses were carried out on all datasets separately and the complete mitogenome in MrBayes v3.2.7 (*Ronquist et al., 2012*) with partitions and models included as per the PartitionFinder results (Table 3). For each of the 22 datasets (13 separate coding genes, two ribosomal genes, *16S* fragment, *COI* fragment concatenated *16S* and *ND2* alignment, concatenated *ND2* and *ND5* alignment, concatenated protein-coding genes, complete mitogenome), two separate runs were made, each with four independent

**Table 3** The partitioning scheme and models of evolution used for each partition generated by PartitionFinder for the phylogenetic analyses of *Bradypodion*.

| Partition | Genes | Model |
|---|---|---|
| 1 | *ATP6, ND2, ND5* | GTR + *I* + *G* |
| 2 | *ATP8* | GTR + *I* + *G* |
| 3 | *COI* | GTR + *I* + *G* |
| 4 | *COII, COIII* | GTR + *I* + *G* |
| 5 | *CytB* | GTR + *I* + *G* |
| 6 | *ND1* | GTR + *I* + *G* |
| 7 | *ND3, ND4, ND4l* | GTR + *I* + *G* |
| 8 | *ND6* | GTR + *I* + *G* |
| 9 | *16S* | GTR + *I* + *G* |
| 10 | *12S* | GTR + *I* + *G* |

MCMC chains of 20 million generations, sampling every 1,000 generations. For every run we assessed convergence using the R package RWTY v1.0.2 (*Warren et al., 2017*) with sampling considered adequate when all parameters have ESS values >200. We also made use of ASTRAL version 5.7.8 (*Zhang et al., 2020*) to reconcile the mitogenome tree from each of the previously estimated single mito gene trees. ASTRAL makes use of the multispecies coalescent and quartet scores to estimate the best-supported species tree assuming a model of gene tree discordance. This assumption is not necessarily relevant when estimating the mitogenome tree because the mitogenome is a single non-recombining locus and therefore each mitochondrial gene is expected to share a single coalescent history. Nonetheless, we performed this analysis as an additional measure of gene tree congruence.

Finally, gene and site concordance factors were calculated by computing individual gene trees and comparing them to the best-supported merged-partitioned mitogenome tree in IQ-TREE v2.0.3 (*Minh et al., 2020b*). Concordance factors are typically used to measure topological discordance among independent loci but here all loci share a single history due to a lack of recombination (*Minh et al., 2020a*). Therefore, we used concordance factors to measure topological differences in resolution rather than independent history.

## Topological congruency and phylogenetic resolution

To assess topological congruency between different gene trees and the complete mitogenome tree, Bayesian consensus trees for each of the 22 datasets were imported into R using the package Ape v5.6-2 (*Paradis, Claude & Strimmer, 2004*). Using the function tree.dist.matrix in the package RWTY, Robinson-Fould (RF) topological distances between trees inferred from different gene regions were calculated pairwise to produce a distance matrix of topological distances. We used the nj function in Ape to generate a neighbour-joining dendrogram from this topological distance matrix. To visualize if topological reliability relates to the length of the gene, we constructed a broken *x*-axis scatterplot in R, using the package ggbreak v0.1.1 (*Xu et al., 2021*) and the gap.plot function, of RF distance from the complete mitogenome against gene length (in 100 bp) for each gene as well as our different concatenated partitions. We also carried out Shimodaira-Hasegawa (SH)

topology tests in IQ-TREE as an additional measure of topological congruency among gene trees (*Shimodaira & Hasegawa, 1999*).

## Saturation

To ascertain whether topological discordance across different genes might be due to saturation, we estimated saturation for each gene separately and the complete mitogenome by plotting uncorrected genetic p-distances against model corrected p-distances for each gene (*Everson, 2015*; *Philippe et al., 1994*) using the package Ape and the dist and dist.correct functions. The difference between these estimates is an approximation of the unobserved mutations hidden by multiple mutations occurring at a site along a branch in our tree. Due to an abundance of gaps in the ribosomal RNAs, which would be ignored when calculating model-corrected p-distances and thereby inflate the slope of the linear regression, we did not estimate saturation plots for these genes. In these saturation plots, the slope of the linear regression is indicative of the degree of saturation, the lower the slope of the linear regression, the more saturated the gene is assumed to be. For this reason, we tabulated the slope of the saturation linear regression for each gene along with other metrics relevant to topology: number of polytomies, mean node posterior probability (calculated based only on the supported nodes within the phylogeny), number of fully bifurcating nodes, RF distance from mitogenome, number of parsimony informative sites, gene length, and standardized information content (SIC, see *Macey et al., 2004*; Table 4).

## Explaining RF distance

In addition to comparing the topology of each gene tree to that of the complete mitogenome, we summed the number of polytomies, mean posterior probability support across the tree, and number of fully bifurcating nodes for each gene tree and partition in our dataset and tabulated these metrics together with gene/alignment length, number of parsimony informative sites, saturation and RF-distance. We sought to assess which of these metrics best explains RF distance. We therefore ran an analysis of variance (ANOVA) on the data tabulated above using the aov function in the stats package in R, omitting number of polytomies, number of fully bifurcating nodes, and mean posterior probability because these are all intrinsically linked to the topology of the tree and therefore to RF distance.

# RESULTS

## Comparison with other published phylogenies

Our complete mitogenome topology and our *ND2-16S* are largely congruent with published dwarf chameleon phylogenies constructed using concatenated *16S* and *ND2* gene fragments (*Tolley et al., 2004*; *Tolley, Chase & Forest, 2008*; *Tolley, Tilbury & Burger, 2022*). However, we find that our phylogeny based on *ND2* alone is more robust and has a topology more similar to our complete mitogenome phylogeny. Despite the similarities between the complete mitogenome phylogeny and the previously published phylogenies, a few differences are apparent. Our full mitogenome topology places *B. setaroi* and *B. caffrum* as sister taxa although the relationship lacks node support (PP = 0.71; BS = 47) and therefore cannot be considered a compelling difference. More notably is that the full mitogenome

phylogeny supports *B. ngomeense* as being nested within the *B. transvaalense* clade (PP = 0.99; BS = 92) instead of sister to that species. This could be due to our better geographic coverage for *B. transvaalense* than in previous studies (*Tolley et al., 2004*; *Tolley, Chase & Forest, 2008*) allowing for a more detailed assessment of these relationships. However, none of our other (single gene or concatenated) analyses showed *B. ngomeense* to be nested within *B. transvaalense*, all of which are similar to previously published topologies, and also show node support for the monophyly of the two species. This poses the question as to whether the full mitochondrial dataset has allowed for better resolution within this particular lineage, or whether certain mito-genes have experienced directional selection, providing differing topological outcomes even within the mitochondrion.

## Phylogeny

To assess if the mitogenome phylogeny can be reconciled from single mitochondrial genes, we estimated Bayesian and maximum likelihood phylogenies for the entire mitogenome as well for each single gene and various combinations of genes (all single gene trees have been archived on the Harvard Dataverse: 10.7910/DVN/RY1RQI). We found some variation in the mean node support and number of fully bifurcating nodes between the whole mitogenome tree and the single gene trees (Table 4). For example, the whole mitogenome inferred tree yielded 50 fully bifurcating nodes with a mean posterior probability support of 0.98 while a short fragment of *16S* only yielded 13 fully bifurcating nodes with a mean posterior probability support of 0.86 (Table 4).

The complete mitogenome consensus phylogenies (Bayesian and maximum likelihood) recovered *Bradypodion* as a well-supported monophyletic clade with most internal nodes well-supported both in terms of bootstrap and posterior probability support (Fig. 1). The coalescent phylogeny estimated using ASTRAL (Fig. S1) was very similar to the Bayesian and maximum likelihood phylogenies except that the coalescent phylogeny does not recover *B. caffrum* and *B. setaroi* as monophyletic sister taxa, but this is not particularly surprising given that the sister-taxon relationship of *B. caffrum* and *B. setaroi* received very low support in the Bayesian and maximum likelihood phylogenies (PP = 0.71; BS = 47). Gene concordance factors reveal that while most nodes in the complete mitogenome consensus phylogeny are well-supported (PP ≥ 0.95; BS ≥ 75), some of these nodes do not show concordance across all gene trees (Fig. 1). For instance, the node separating the *B. caffrum-B. setaroi* clade from the *B. caeruleogula-B. transvaalense* clade is well supported by both Bayesian and maximum likelihood methods, but it has a gene concordance factor of only 33.3, meaning that only 33.3% of the gene trees in the mitogenome support that node (Fig. 1). The placement of *B. caeruleogula*, while well-supported in both bootstrap and posterior probability support, is only supported by 53.3% of gene trees. In both of these instances, the site concordance factors are also low (34.09% and 38.58% respectively (Fig. S2). These low concordance factors might reflect a lack of phylogenetically informative information in some of the genes, hindering phylogenetic estimation for these genes. Typically, when site and gene concordance factors agree (which they tend to in our analyses), this suggests genuine topological conflict between genes. However, given that the

**Table 4  Measures of topological congruency between gene trees.** Number of polytomies, saturation linear regression slope, mean posterior probability support, number of fully bifurcating nodes, alignment length, number of parsimony informative sites, SIC, RF distance from the complete mitogenome for each gene and partition, and Shimodaira-Hasegawa (SH) topology test p-value from the phylogenetic analyses of *Bradypodion*.

| Gene/partition | Polytomies[*] | Saturation regression slope | Mean node posterior probability | Fully bifurcating nodes | Alignment length (bp) | Parsimony informative sites | SIC[#] | RF distance[†] | SH test p-value |
|---|---|---|---|---|---|---|---|---|---|
| mitogenome | 0 | 0.785 | 0.978 | 50 | 14,028 | 5,576 | 397.49 | 0 | 1 |
| protein coding genes | | | | | | | | | |
| *ND1* | 22 | 0.788 | 0.945 | 30 | 966 | 377 | 390.27 | 30 | 0 |
| *ND2* | 17 | 0.756 | 0.946 | 38 | 1,059 | 469 | 442.87 | 13 | 0.007 |
| *ND3* | 25 | 0.761 | 0.858 | 27 | 351 | 148 | 421.65 | 34 | 0 |
| *ND4* | 20 | 0.769 | 0.953 | 34 | 1,362 | 588 | 431.72 | 20 | 0 |
| *ND4l* | 38 | 0.733 | 0.859 | 19 | 289 | 127 | 439.45 | 36 | 0 |
| *ND5* | 11 | 0.743 | 0.971 | 41 | 1,840 | 829 | 450.54 | 13 | 0.418 |
| *ND6* | 30 | 0.691 | 0.89 | 25 | 534 | 231 | 432.58 | 26 | 0 |
| *ATP6* | 27 | 0.702 | 0.914 | 28 | 690 | 304 | 440.57 | 26 | 0.010 |
| *ATP8* | 37 | 0.639 | 0.862 | 19 | 168 | 102 | 607.14 | 33 | 0 |
| *COI* | 15 | 0.823 | 0.946 | 38 | 1,549 | 520 | 335.7 | 19 | 0.038 |
| *COII* | 33 | 0.819 | 0.903 | 24 | 684 | 216 | 315.79 | 31 | 0 |
| *COIII* | 18 | 0.796 | 0.923 | 36 | 785 | 295 | 375.8 | 18 | 0.223 |
| *Cyt-b* | 12 | 0.778 | 0.909 | 38 | 1,141 | 451 | 395.27 | 33 | 0 |
| RNA genes | | | | | | | | | |
| *16S* | 15 | – | 0.941 | 34 | 1,653 | 582 | 352.09 | 31 | 0 |
| *12S* | 32 | – | 0.898 | 24 | 958 | 337 | 351.77 | 32 | 0 |
| concatenated genes | | | | | | | | | |
| *rRNA (16S/12S)* | 21 | – | 0.95 | 34 | 2,638 | 919 | 348.37 | 27 | 0 |
| *ND2-16S* | 19 | – | 0.95 | 35 | 1,520 | 579 | 380.92 | 21 | 0.012 |
| *ND2-ND5* | 3 | 0.747 | 0.983 | 46 | 2,899 | 1,298 | 447.74 | 9 | 0.963 |
| Protein coding (PCG) | 0 | 0.778 | 0.958 | 50 | 11,417 | 4,657 | 407.9 | 8 | 0.943 |
| gene fragments | | | | | | | | | |
| Short *16S* | 44 | – | 0.857 | 13 | 461 | 110 | 238.61 | 39 | 0 |
| Short *COI* | 26 | 0.807 | 0.922 | 26 | 624 | 231 | 370.19 | 26 | 0 |

**Notes.**

*In this context, number of polytomies refers to the number of branches directly emanating from polytomies within the phylogeny

†Robinson-Fould distance from mitogenome phylogeny

#SIC stands for standardized information content and is calculated as the ratio of parsimony informative sites to the total alignment length per kilobase (Macey et al., 2004).

mitochondrion is a single non-recombining locus, this conflict is more likely an artefact of certain genes lacking the phylogenetic information to converge on the true topology.

## Analysis of topological congruence and phylogenetic resolution

We found varying degrees of topological inconsistency across phylogenies inferred by different genes within the mitogenome (Fig. 2). The dendrogram of RF topological distances shows that *ND2, ND5, COIII,* the concatenated PCG alignment, the concatenated *ND2*-16S and the concatenated *ND2-ND5* all cluster with the tree inferred from the complete mitogenome, suggesting that the phylogenies inferred from these single genes or concatenations most closely resemble that of the complete mitogenome. However, the

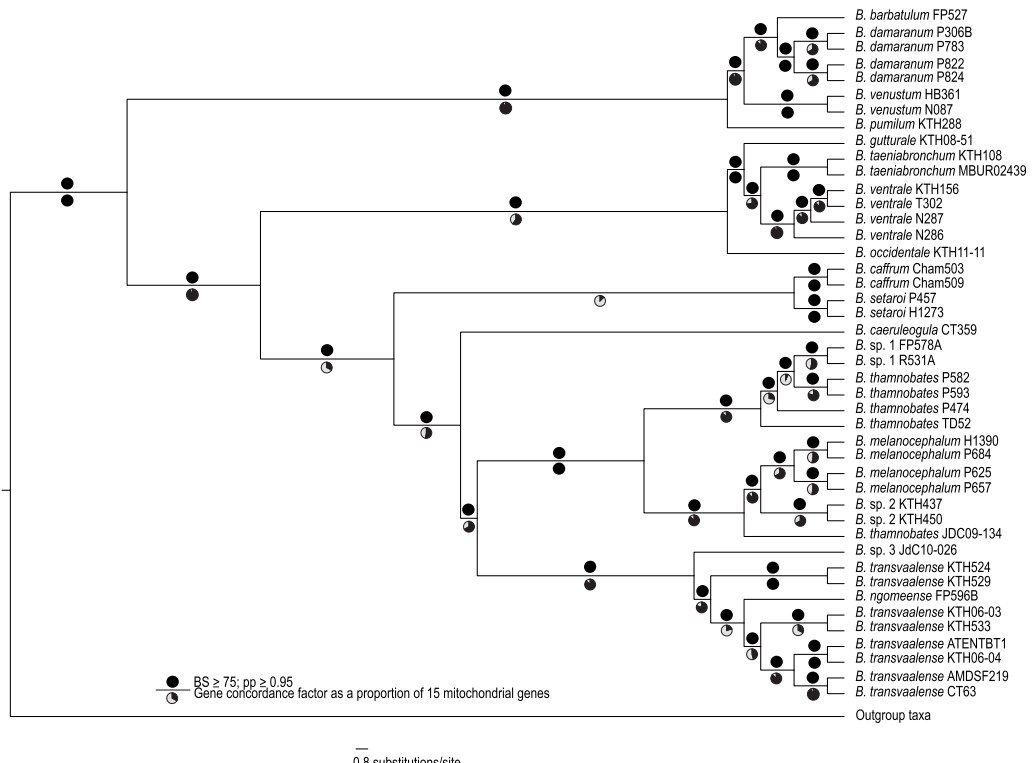

**Figure 1** **A best-supported Bayesian phylogeny for *Bradypodion* based on the complete mitogenome.**
Black circles above branches represent nodes with ≥ 0.95 posterior probability and ≥ 75% bootstrap support. Gene concordance factors are indicated below branches in proportional pie charts. These pie charts represent the proportion of genes (out of 15) that support a given node within the phylogeny.

concatenated *ND2-16S* alignment is more distant from the complete mitogenome than the *ND2* gene on its own, despite the concatenated alignment consisting of more characters. Sister to this cluster is a grouping of *ND4* and *COI*, and just outside this cluster is *ND6*. More distant from the complete mitogenome are the ribosomal RNAs and *ND3*, *Cyt-b*, *COII*, *ND4l*, *ATP6*, *ATP8*, the shortened *16S* fragment, and *ND1*.

It is apparent that RF distance from the complete mitogenome tree shows an inverse correspondence with gene length; however, there is considerable variation which indicates that some genes hold more information than others (Fig. 3). For instance, our longest alignment was for the combined protein-coding genes which also had the lowest RF distance from the mitogenome tree, while the shortest alignment was the *ATP8* gene which had one of the highest RF distances from the mitogenome. *ND2* had one of the lowest RF distances from the mitogenome, even lower than the concatenated *ND2-16S* alignment. Furthermore, the concatenated rRNA alignment was longer than all individual genes and yet still had a relatively high RF distance from the mitogenome, so phylogenetic informativeness does not correlate highly with gene fragment length. Nonetheless, concatenating two of the most reliable single genes, *ND2* and *ND5* produced a topology even closer to the complete mitogenome than either of these genes on their own.

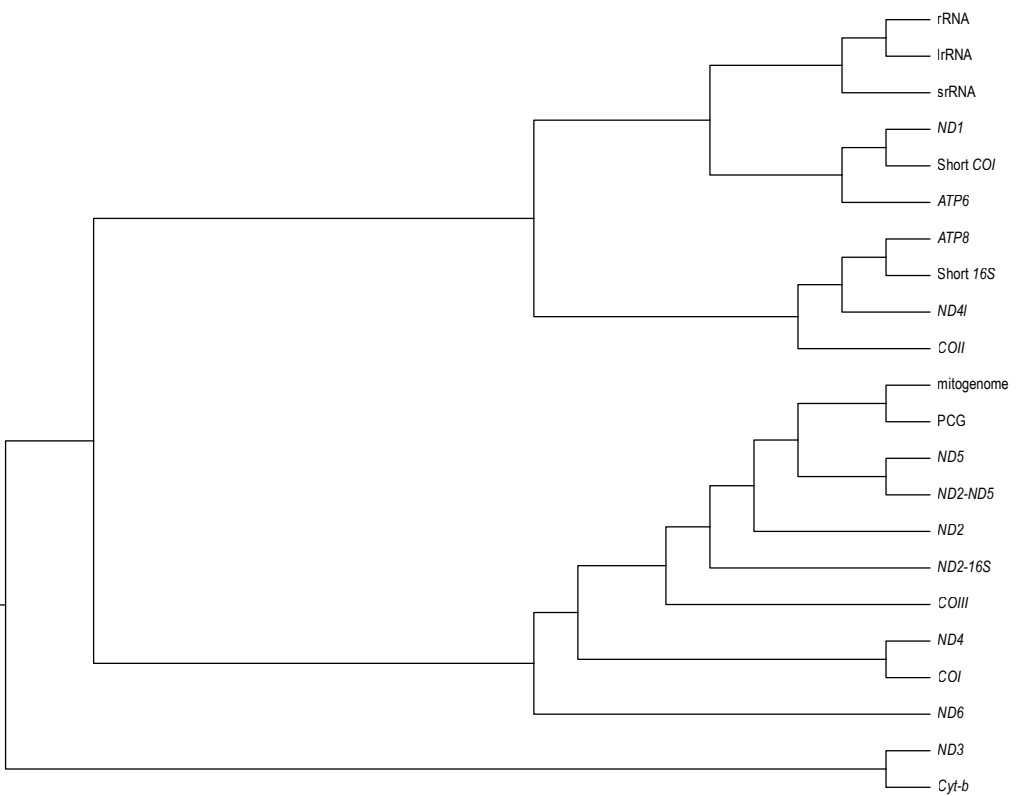

**Figure 2** **A neighbour-joining dendrogram for Robinson-Fould distances between *Bradypodion* topologies inferred for different genes in the mitogenome.** PCG, protein coding genes; short *16S*, short fragment of *16S*; short *COI*, short fragment of *COI*; *ND2-16S*, concatenated alignment of *ND2* and a short fragment of *16S*, *ND2-ND5*, concatenated alignment of *ND2* and *ND5*.

While the SH-tests also showed *ND5* to be the best single gene proxy for the complete mitogenome, there were a few discrepancies between the RF-distances from the complete mitogenome and the results of the SH-tests (Table 4). The SH-tests supported *COIII* as the second-best proxy for the mitogenome, with no other single genes being congruent to the complete mitogenome. Furthermore, the SH-tests supported the concatenated alignment of *ND2* and *ND5* as a better proxy for the complete mitogenome than the PCG alignment.

As might be expected, the complete *Bradypodion* mitogenome phylogeny scores best with the fewest polytomies, the most fully bifurcating nodes, and the highest mean posterior probability support and an SH-test *p*-value of 1. Based on these same criteria, the short *16S* fragment scored lowest with the most polytomies, lowest mean posterior probability support, and the fewest fully bifurcating nodes. It should be noted that mean posterior probability node support is calculated based only on the supported nodes within the phylogeny. If there is an unsupported polytomy, this is left out of the calculation, so these values should be consulted in conjunction with the number of polytomies and

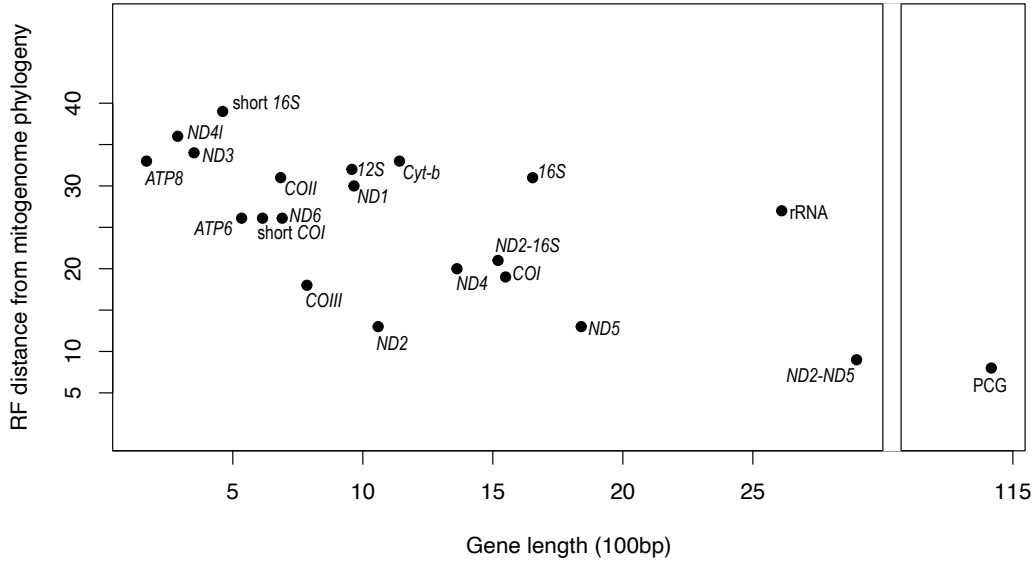

**Figure 3** **Gene fragment length (in 100 bp) *versus* Robinson-Fould distance from the complete mitogenome phylogeny for each mitochondrial gene fragment/partition for *Bradypodion*.** PCG, concatenated alignment of all protein coding genes; short *16S*, short fragment of *16S*, short; *COI*, short fragment of *COI*; rRNA, concatenated alignment of *12S* and *16S*; ND2-16S, concatenated alignment of *ND2* and a short fragment of *16S*, ND2-ND5, concatenated alignment of *ND2* and *ND5*.

number of fully bifurcating nodes. The protein coding gene alignment, *ND2*, *ND5*, and the concatenated *ND2* and *ND5* alignment performed best across most of these metrics.

It is apparent from our ANOVA that saturation is not driving topological discordance within the mitogenome ($p = 0.59$; $F = 0.30$). This is probably due to strong structural constraints on the evolution of protein-coding genes within the mitochondrion, constricting the degree to which third position bases mutate. We did, however, find RF distance from the mitogenome to correlate significantly with gene/alignment length ($p = 0.006$; $F = 11.36$), and we found a significant interaction between the number of parsimony-informative sites and total gene alignment length ($p = 0.02$; $F = 7.36$).

## DISCUSSION

Our findings suggest that, if chosen carefully, single mitochondrial protein-coding gene trees can be nearly as informative as full mitogenome datasets that are produced by expensive new technologies. For *Bradypodion*, we find that certain mitochondrial genes produce topologies that are essentially identical to that of the entire mitogenome and have almost equally strong support at most nodes. Discrepancies are largely limited to nodes that lack support across all datasets including the full mitogenome dataset. The *ND2* and *ND5* genes produced topologies that were the most similar to the mitogenome, as well as to the concatenated protein coding genes phylogeny. These topologies were further improved (at least in terms of RF distance from the complete mitogenome) when *ND2* and *ND5* were concatenated. Overall, Bayesian and likelihood methods recovered the same topology and

similar node support regardless of dataset. However, one node (*B. caffrum-B. setaroi*) in the complete mitogenome phylogeny lacked support regardless of dataset used or analytical approach, and that node has never been recovered in previous phylogenies (*Tolley et al., 2004*; *Tolley, Chase & Forest, 2008*; *Tolley, Tilbury & Burger, 2022*).

While our findings demonstrate that datasets with a larger number of base pairs provide improved mean node support, the differences appear to be fairly subtle. Most mitogenome-inferred, species-level relationships within *Bradypodion* are also well reconciled using single gene fragments. In particular, single genes *ND5*, *ND2*, *COI*, *COII*, and *ND4* resolved all major relationships. The phylogenetic utility of these markers is supported by multiple different analyses (*i.e.,* RF distances, number of polytomies, mean posterior probability support, and number of fully bifurcating nodes). The finding that certain single mito-genes reflect the entire mitogenome adequately is encouraging given that markers such as *ND2*, *ND4*, *COI* as well as *ND2-16S* have been widely used in reptile systematics over the past few decades (*Bates et al., 2013*; *Laopichienpong et al., 2016*; *Macey, Schulte & Larson, 2000*; *Main et al., 2022*; *Nagy et al., 2012*; *Taft, Maritz & Tolley, 2022*).

The topological similarities between our complete mitogenome phylogeny and previous phylogenies that included *ND2* (*Tolley, Tilbury & Burger, 2022*) echo the sentiment that *ND2* is a good proxy for the complete mitogenome. However, we find that the resolution of *ND2* is weakened by its concatenation with a short fragment of *16S*. This suggests that, at least for *Bradypodion*, short of using genomic data, it might be more appropriate to use only *ND2* in small-scale phylogenetic analyses going forward, or to combine *ND2* with another gene with a low RF score, such as *ND5*. Overall, the shortened fragment of *16S* commonly used in reptile and amphibian phylogenetics is probably a poor marker for phylogenetic reconstruction, and this finding is not unique. A previous study similarly found that, for amphibians, short *16S* yields poor and erratic phylogenetic estimations and that these can be improved by using longer fragments of *16S* (*Chan et al., 2022*). Correspondingly, we also found that the complete 1,653 bp *16S* gene performed notably better than the shortened fragment (ca. 461 bp), in terms of RF distance from the mitogenome, number of polytomies and number of fully bifurcating nodes (Table 4). However, the full *16S* gene still performed poorly in recovering the full mitogenome topology, as compared to coding genes, *i.e., ND2*, *ND4*, *ND5*, *COI*, and *COII*. Considering the popularity of *16S* in herpetological systematics (*Alrefaei, 2022*; *Bates et al., 2013*; *Hay et al., 1995*; *Hertwig, de Sá & Haas, 2004*; *Lamb & Bauer, 2002*; *Main, Van Vuuren & Tolley, 2018*; *Main et al., 2022*; *Vences et al., 2004*; *Vences et al., 2005*; *Welton et al., 2010*), the inadequacy of *16S* at reconciling the complete mitogenome tree is concerning, particularly when used as a single gene marker for phylogenetic estimation. While most of these studies make use of other makers as well, in most cases *ND2* or *ND4*, our results suggest that the inclusion of both short and long fragment *16S* for phylogenetic analysis confounds the signal of the coding genes, and thus might reduce the resolution of the topologies. The commonly used *COI* barcoding fragment (*Hebert et al., 2003*) did not recover as much resolution as the complete *COI* gene but it still outperformed the shortened *16S* fragment. We advocate that both effort and expense could be saved by avoiding the use of *16S*. We acknowledge that our study has used only one reptile genus as a model, but it is likely that these inferences

apply more widely than for only *Bradypodion*. However, it would be useful to confirm these results by carrying out additional phylogenetic mitogenome analyses using other taxa and comparing topologies amongst full genomes and single genes.

Gene concordance factors revealed that certain clades are more robustly supported across the mitogenome than others. For instance, over 90% of the mitochondrial gene partitions supported the clade containing *B. pumilum*, *B. barbatulum*, *B. venustum*, and *B. damaranum* as monophyletic; however, fewer than 15% of gene partitions support *B. setaroi* and *B. caffrum* as sister taxa. At the species level, the sister relationship of *B. setaroi* and *B. caffrum* has not been recovered in previous analyses (*Tolley et al., 2004*; *Tolley, Chase & Forest, 2008*), although this node also lacks node support in the present analysis. The relationship between *B. transvaalense* and *B. ngomeense*, albeit with less extensive geographic coverage than we have in our analyses, has historically been considered closely related (*Tilbury & Tolley, 2009*). Our full mitogenome phylogeny recovers *B. ngomeense* as paraphyletic with *B. transvaalense*. Further, the placement of *B. caeruleogula* has been uncertain, as it has either been part of a deep polytomy or placed with weak node support as sister to the *B. thamnobates* clade (*Tilbury & Tolley, 2009*). Both the placements of *B. caeruleogula* and *B. occidentale* have now been resolved with good support (PP $\geq$ 0.95; BS $\geq$ 75) by the complete mitogenome phylogeny.

While our results demonstrate that certain single mitochondrial genes are effective proxies for reconciling the mitogenome phylogeny two caveats should be mentioned. Firstly, phylogenetic trees are merely informed estimates, trees that are well-supported and well-resolved are not necessarily reflecting the evolutionary history of the taxa, they are just our best estimates given the data. We used the full mitogenome as our gold standard with which to compare other genes because it holds the most data, and we therefore assume it to have the best priors to inform phylogenetic estimation. We did not, however, set out to test for phylogenetic *accuracy*, we only sought to test whether single or concatenated mito-genes could reconcile the same mitogenome phylogeny as the full mitogenome. Secondly, even if we assume the tree to be *accurate*, this does not mean that the mitogenome tree necessarily reflects the species tree. Mito-nuclear discordance, due to factors like assortative mating, sex-linked selection, asymmetric introgression, and incomplete lineage sorting, is well documented for other taxa (*Després, 2019*; *Fossøy et al., 2016*; *Toews & Brelsford, 2012*; *Tóth et al., 2017*; *Wendt et al., 2022*). To assess whether this is the case for *Bradypodion*, a whole genome phylogenetic analysis is required. Fortunately, the mitogenome phylogeny serves as an important starting resource for future mito-nuclear comparisons. While our results are applicable to *Bradypodion* and might be more generally applicable to other reptile taxa, we still emphasize care when interpreting evolutionarily relationships based on single/concatenated mito-genes only. That is, for a fuller assessment of the history of a taxonomic group, nuclear genes that mutate at different rates must be included. In addition, new methods have allowed for genome scanning and these have proven very useful in the discovery of mito-nuclear discordance, and even speciation in the presence of gene flow with introgression (see *e.g.*, *Kunerth et al., 2022*; *Mikkelsen & Weir, 2023*; *Perea et al., 2016*; *Wright et al., 2022*).

## CONCLUSIONS

Our results provide compelling evidence for the efficacy of single mito-genes such as *ND5*, *ND2*, *COI, COII*, and *ND4* (or concatenation of some) for reconciling the mitogenome phylogeny within the genus *Bradypodion*. Whether mitochondria themselves are adequate proxies for evolutionary history remains to be demonstrated. Nevertheless, these results are promising and suggest that apart from relying on *16S* as a marker of choice, there is still utility in producing single/few gene sequence phylogenies. Furthermore, for researchers that are not yet able to take advantage of whole genome sequencing, readily sequenced genes like *ND2, ND5* or *ND4* appear to be reasonable proxies for phylogenetic estimation.

## ACKNOWLEDGEMENTS

Thanks to Jonathan Losos and Graham Alexander for assisting in generating the funding for this project, and to the many field workers who assisted with sample collections over the years. DNA library preparation and sequencing were performed by The Bauer Core Facility at Harvard University.

### Funding

This work has been funded by the National Research Foundation of South Africa (grant number 136381) and the National Science Foundation (award numbers 1927194 and 2152059). The funders had no role in study design, data collection and analysis, decision to publish, or preparation of the manuscript.

### Grant Disclosures

The following grant information was disclosed by the authors:
The National Research Foundation of South Africa: 136381.
The National Science Foundation: 1927194, 2152059.

### Competing Interests

The authors declare there are no competing interests.

### Author Contributions

- Devon C. Main conceived and designed the experiments, performed the experiments, analyzed the data, prepared figures and/or tables, authored or reviewed drafts of the article, and approved the final draft.
- Jody M. Taft conceived and designed the experiments, performed the experiments, analyzed the data, authored or reviewed drafts of the article, and approved the final draft.
- Anthony J. Geneva conceived and designed the experiments, performed the experiments, analyzed the data, prepared figures and/or tables, authored or reviewed drafts of the article, acquired funding, and approved the final draft.

- Bettine Jansen van Vuuren conceived and designed the experiments, authored or reviewed drafts of the article, acquired funding, and approved the final draft.
- Krystal A. Tolley conceived and designed the experiments, authored or reviewed drafts of the article, acquired funding, and approved the final draft.

## Animal Ethics

The following information was supplied relating to ethical approvals (*i.e.,* approving body and any reference numbers):

Ethics clearances were obtained from the South African National Biodiversity Institute (003/2011, 001/2013, 001/2014, 0001/2015), University of Johannesburg (2019-10-10), and University of the Witwatersrand: (2019/10/56/B).

## Field Study Permissions

The following information was supplied relating to field study approvals (i.e., approving body and any reference numbers):

Samples for this study were collected under permits from the provinces of Eastern Cape, KwaZulu-Natal, Limpopo, Mpumalanga, Northern Cape, Western Cape (018-CPM403-00001, 0092-CPM401-00006, AAA004-00107-0035, CPM-333-00002, CRO 3/19CR, CRO 35/15CR, CRO 36/15CR, CRO 32/20CR, CRO 33/20CR, FAUNA 110/2011, OP 4758/2010, OP 4596/2010, OP 2635, OP 1259/2014, KZN 1647/2009, MPB.5299, MPB.5371, MPB.5544, MPB.5604, CN44-59-5795, RSH 24/2021, WRO 41/03WR; WRO 15/03WR).

Subsamples were exported to the USA under CITES permits (#142667, 206199, 257511, 260568 and 260570) issued by the South African Department of Fisheries, Forestry and Environment), with remaining tissue samples accessioned at the South African National Wildlife Biobank, Pretoria.

## DNA Deposition

The following information was supplied regarding the deposition of DNA sequences:

The 660 sequences are available at GenBank (Table S1).

## Data Availability

The alignments, phylogenetic trees, saturation plots, and R scripts are available on the Harvard Dataverse: Main, Devon, 2024, ''Replication Data for: The efficacy of single mitochondrial genes at reconciling the complete mitogenome phylogeny –A case study on dwarf chameleons'', https://doi.org/10.7910/DVN/RY1RQI, Harvard Dataverse, V1, UNF:6:HyZpnEDX2LYCdneRXAsZpA== [fileUNF].

## Supplemental Information

Supplemental information for this article can be found online at http://dx.doi.org/10.7717/peerj.17076#supplemental-information.

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
