# Peer review of "The efficacy of single mitochondrial genes at reconciling the complete mitogenome phylogeny—a case study on dwarf chameleons"

_PeerJ, doi:10.7717/peerj.17076_

## Round 0.1 · original submission · Major Revisions

Dear Authors,

Upon careful consideration of feedback from four reviewers, I am pleased to convey their unanimous appreciation for the exceptional quality of your manuscript, recognizing its significance within the scientific community. However, they have put forth constructive suggestions for refinement, particularly in the domain of Experimental Design. Emphasizing the importance of avoiding resulting generalization, both in the title and the content, is crucial. It is recognized that the identified pattern in this manuscript may not universally apply across taxonomic groups, therefore major corrections are necessary.

Sincerely,

Armando Sunny

Reviewer 1 ·

Basic reporting

Good background and knowledge gap. The authors place the research into context, why mitochondrial genes are used in phylogenetics, how this does not necessarily match an phylogeny based on the nuclear genome, the increased affordability of sequencing mitogenomes, that low income economies might not have the resources to sequence mitogenomes and that there is already an abundance of single gene data in online repositories that could be used in future research.

Its acknowledged that complex genealogies may need more than single genes to resolve phylogenetic questions.

Clear identification of the gap in the literature – which is that single genes can infer the same or very similar phylogeny to the full mitogenome and which genes are best for this for reptiles has not yet been identified. Clear hypothesis.

Experimental design

Sound methodology with enough detail to replicate.

Validity of the findings

They are not trying to claim more than what their results show in the discussion and does open up for future research questions. All sampling and ethics permits have been obtained and data will be deposited in GenBank.

Additional comments

‘More is not always better: the efficacy of single mitochondrial genes at reconciling the complete mitogenome phylogeny’

Don’t have any major issues with the study as it is, just a few minor points below for clarification.

Introduction

• Last sentence in the introduction, does not seem part of the hypothesis and reads more like the end of the discussion/conclusions. Also does not fit well with the rest of the introduction which seems to favour using single genes where possible.

Materials & methods
• Under phylogenetic analyses this sentence ‘we created 22 alignments for subsequent phylogenetic analyses of single genes and for a few specific combinations of genes that reflect typical concatenated datasets used previously’ – only reference is Hebert et al. 2003 after ‘short fragment of COI’ – please include a few more references showing how these concatenated genes are commonly used in herpetological phylogenetic literature.
• “We created 16 single locus alignments - one each of 14 protein-coding genes as well as for the genes coding for the large and small ribosomal subunits” -> this should be 15 single locus alignments (13 protein coding genes and 2 ribosomal subunits) + 7 additional alignments = 22 total alignments
• “For each of the 21 datasets (14 separate coding genes, 2 ribosomal genes, 16S fragment, concatenated 16S and ND2 alignment, concatenated ND2 and 182 ND5 alignment, concatenated protein-coding genes, complete mitogenome)” – why have you excluded short CoI and rRNA alignments here? Should be 13 separate coding genes, 2 ribosomal and then 7 other alignments for a total of 22 to match the rest of the paper

Results
• Line 244-245: ‘We found some variation in the average 244 node support and number of fully bifurcating nodes between the whole mitogenome tree and the 245 single gene trees (Table 1).’ -> should this be Figure 1?
• Line 246-248: ‘For example, the whole mitogenome inferred tree yielded 50 fully 246 bifurcating nodes with an average posterior probability support of 0.978 while a short fragment 247 of 16S only yielded 13 fully bifurcating nodes with an average posterior probability support of 248 0.857 (Table 1).’ -> should this be table 4?
• Lines 289-297 reads more like it should be in the methodology section instead of the results section
• Figure 1 should say 15 not 16 genes

Discussion
• Line 371 – says table 1 but this should be table 4
• Line 381 fragment(Herbert needs a space
• Line 397 2022) . -> remove gap between bracket and fullstop
Conclusions
• Should mention that this is for Bradypodion taxons only

Reviewer 2 ·

Basic reporting

Dear Authors,
I was very happy to act as a reviewer for this interesting study, entitled "More is not always better: the efficacy of single mitochondrial genes at reconciling the complete mitogenome phylogeny". I agree with the general conclusions of the manuscript and I wish to thank you for providing such a case study.
I have a big concern, which cannot be fixed, but should be addressed in the Discussion in my opinion: some single-gene phylogenies are useful and informative, and probably as accurate as large, complete-mitogenome phylogenies; however, you needed complete-mitogenome phylogenies as a benchmark to spot out informative single genes. This may hamper the rationale that single genes can be directly used as proxies in phylogenetic studies, since they are not a priori known, and may vary from those useful for dwarf chameleons. At least, I suggest to insert in the title of the paper something like "a case study on dwarf chameleons".
Generally speaking, the manuscript is clear, but I think that it is a bit confusing, and it would be difficult to reproduce it in my opinion, since many details are lacking and other are ill-defined. I provide a list of minor concern that should be fixed prior to publication in my opinion, hoping to help you to improve the manuscript further.
Best wishes!

Experimental design

The experimental design is generally clear; however, as stated above, more details are needed somewhere: please refer to the attached PDF.

Validity of the findings

I globally agree with the conclusions of the manuscript, albeit a have a major concern on the generalization, as I explained above. Since there is basically no novelty in the phylogenetic outcome of the paper, the relevance of the main rationale fo the article (the soundness of single-gene phylogenies) should be thoroughly discussed.

Annotated reviews are not available for download in order to protect the identity of reviewers who chose to remain anonymous.

Reviewer 3 ·

Basic reporting

The molecular phylogenetics survey conducted by Main et al. was skillfully written and included an ample amount of relevant background information pertaining to their study system. However, the title of the paper is misleading and the reviewer believes the generalization does not hold across taxonomic groups. It may hold at the genus-level, but not necessarily at the family, superfamily, infraorder, or order levels. The reviewer recommends that the author include that the taxonomic level their title applies to is the genus-level.

Experimental design

During the present era of phylogenomics, the reviewer determines that the study hypothesis (Lines 103-111) is outdated no longer a matter of debate. Rather, the authors should reformulate their hypothesis towards understanding the causality of phylogenetic discordance with single vs multiple gene datasets.

The remarks in Lines 90-96, which are cited as a justification, are not completely correct, indicating an important oversight on the part of the author. Phylogenomics techniques have progressed to utilize the multi-species coalescent model (MSCM), which enables the consideration of the impact of gene tree discordance on species tree inference. An example of this is the Accurate Species TRee ALgorithm, commonly referred to as ASTRAL, as described by Zhang et al. (2018; https://doi.org/10.1186/s12859-018-2129-y). ASTRAL presents a prominent method for deducing species trees based on gene trees, taking into consideration the phenomenon of incomplete lineage sorting. Furthermore, instead of using the gene tree-based ASTRAL methodology, one can use the site-based SVDQuartets method (Chifman and Kubatko, 2015) to assess the impact of gene tree conflict on species tree inference using the multispecies coalescent model. Moreover, to reduce the adverse impacts of compositional heterogeneity, site-heterogeneous mixture models (CAT and C10-60) have been developed and are extensively employed in the field of phylogenomics. Therefore, from a methodological standpoint, it is not logical to support the utilization of data from a single gene to resolve phylogenetic relationships.

This section ‘Phylogenetic analyses’ requires more detail regarding the alignment methods use for protein-coding genes (PGCs) and for the genes coding for the large and small ribosomal subunits, respectively. Moreover, clarify that the details for 22 alignments for subsequent phylogenetic analyses can be found in Table 4, although it is not clear how the partitioning scheme was done. The reviewer was expecting a similar approach to Leavitt et al. (2023: https://doi.org/10.1016/j.ympev.2013.02.019) given that evolutionary model-fit is improved by partitioning alignments into relatively homogeneous sets of sites before selecting and optimizing a substitution model for each set independently. A major flaw according to Table 3 that states that partitioning was solely based on gene boundaries. Also, to select the to select the best-fitting partitioning scheme and models of evolution within IQTREE, ModelFinder (Kalyaanamoorthy et al., 2017; https://doi.org/10.1038/nmeth.4285) should have been used, and for selecting the appropriate models for MRBAYES.

Hence, the reviewer doubts that the experimental design employed in the current study well represents the generally utilized methods for determining phylogenetic relationships using multilocus datasets to allow for a fair comparision of single vs multigene phylogeny reconstruction.

Validity of the findings

At present, a robust phylogeneny reconstruction approach is warrented before any conclusions are drawn.

Additional comments

Lines 69-81: The reviewer does not dispute that the economy can be a limiting factor, but this should not be at the expense of robust phylogeny reconstructions. The reviewer firmly believes that is also paramount to consider the evolutionary depth of studied taxa.

Lines 83-85: Also, elasmobranchs, see Naylor et al. (Bulletin of the American Museum of Natural History, 2012(367):1-262 (2012). https://doi.org/10.1206/754.1).

Lines 154-155: Please check and correct accordingly. There are 13 PCGs, no? See Line 161-162. Please clearly demarcate the different datasets as Dataset 1, -2, etc, so it will be easy to follow the results.

Reviewer 4 ·

Basic reporting

The manuscript deals with an interesting and quite current issue: the use of large quantities of data while it might be possible to get the same with smaller efforts.
This is a really interesting point since we are, currently, in the transitional phase from PCR-generated and genomic(NGS)-generated data and, as correctly pointed out by Authors, many are still in the condition to not afford the NGS procedure.
I have mixed feelings about that and about the manuscript which I am going to try to explain.
I am convinced that we have to make our best efforts to move to genomic(NGS) data and leave PCR(+Sanger) for very special situations. On top of this, it is a matter of fact that single-gene phylogeny will always be suspected of being biased, even though there are many examples of single-genes analyses later validated by genome-scale analyses.
Therefore, I am not necessarily in agreement with the main message of this manuscript in the sense that finding a good proxy I don't think it is an improvement as one never know if in unexplored taxa that same proxy could really work. On the other hand, analyses like this could be a mean to validate past works and confirm/reject previous phylogenetic inferences.

Experimental design

From a technical point of view I have only one major concern, and regards the topological congruence between mitochondrial genome topology and single-gene topologies. Authors use RF distance, which is a good tool when dealing with a large dataset. Though, I think the more appropriate way to test congruence is the use of topological test like the approximately unbiased test, the Shimodaira-Hasegawa test and other similar to them. These tests could tell something more about congruence, and eventually also on proxability, as they test compatibility of two different topology. For example, low RF between two topologies does not automatically means they can be seen as compatible.
My suggestion, give the limited set of trees to compare, is to include topology tests along with the current RF analysis, to check whether the two results are similar or not and, possibly, to comment/discuss any difference.

Validity of the findings

Overall, the aim and scope of the manuscript are well-explained, reasonable, the analyses are mostly appropriate (but see above) and conclusions sound (although my concerns remain). However, it is unfortunate that some data do not appear properly addressed: for example, gene concordance factors are not deeply investigated and it could be worth more discussion about that. And the same is about saturation: for example it could be nice to see whether gene concordance factors and site concordance factor agree or not, which could tell more on the role of saturation and gene's phylogenetic informativity.

Additional comments

More specific comments:
line 154-155: 14 PGCs are mentioned, while usually mitochondrial genome host 13 PCGs. Please check or better explain
line 170: Since the use of IQTREE, why not using the integrated ModelFinder?
line 243: I cant access the link. Maybe it is just me but, please, check for link coorectedness
line 252-255: ambiguous writing: the use of the term "resolved" to define node with low concordance factor is misleading, as that term is usually associated to bootstrap/posterior support. I suggest to rephrase, like "...some of these nodes do not show concordance across gene trees (Fig. 1)." or something like this.
line 266: "...the 15 protein-coding genes..." wording is misleading. In the context of this paper, it looks like the mitochondrial genome have 15 PCGs, while it is as usual 13 PCGs. I understand the need to be comprehensive while writing but in some cases it is just better to explicitate (and distinguish) PCGs alignments from concatenated alignments of group of genes (and, BTW, I can't think about ND-16S as PCG alignment since 16S do not code for a protein)

---

## Round 0.2 · accepted · Accept

Dear Authors,

It brings me great pleasure to convey to you the news that your manuscript has undergone substantial enhancements, skillfully incorporating all revisions requested by the reviewers. Consequently, I am thrilled to announce the acceptance of your work for publication.

Warm regards,

Armando Sunny

Reviewer 1 ·

Basic reporting

The whole paper is a lot more clear now after the authors have addressed reviewer concerns and rewritten some sections.

Experimental design

Addition of some extra analyses as per reviewers' requests have added to the paper's experimental design.

Validity of the findings

no comment

Additional comments

The authors have received peer review feedback from 4 reviewers and addressed the concerns and comments of all reviewers very well which has led to a more clearly written manuscript with stronger methodology and discussion. I am happy to accept it as is.

One minor typo is:

L343 on -> no

Reviewer 2 ·

Basic reporting

Dear Authors,
I wish to thank you very much for the revision work that work carried out. I did my best to help in improving your manuscript and I really hope that you also notice the improvement of the paper after the revision. The manuscript is now ready for publication in my opinion.
Best regards!

Experimental design

no comment

Validity of the findings

no comment

Reviewer 3 ·

Basic reporting

no comment

Experimental design

no comment

Validity of the findings

no comment

Additional comments

no comment

Reviewer 4 ·

Basic reporting

In the new version of the manuscript, reporting is improved although I feel like there should ba a language check: unusual wording and phrasing are pervasive in the manuscript

Experimental design

The experimental design is surely improved and more complete

Validity of the findings

Findings are based on results, Authors avoid speculations. Suggested changes have been successfully introduced.

Additional comments

I have to say that, although an interesting exercise, I do not feel this analysis is a game changer nor will be of any help in the future. I do not share the view of Authors and I am convinced that we shall out much efforts in genome-scale analyses and in popularizing their use (including making more accessible to low-income countries or small research group withut significant funding) rather than finding short-cuts. However, these are no reason to reject in PeerJ and I find it legit.